# Beyond Wolfram Syndrome 1: The *WFS1* Gene’s Role in Alzheimer’s Disease and Sleep Disorders

**DOI:** 10.3390/biom14111389

**Published:** 2024-10-31

**Authors:** Valerio Caruso, Luciana Rigoli

**Affiliations:** 1Department of Neuroscience, Psychiatric Section, Azienda Ospedaliera Universitaria Pisana (A.U.O.P.), 56126 Pisa, Italy; valeriocaruso79@gmail.com; 2Department of Human Pathology of Adulthood and Childhood G. Barresi, University of Messina, 98125 Messina, Italy

**Keywords:** *WFS1* gene, Wolfram syndrome 1, Alzheimer’s disease, neuropsychiatric disorders

## Abstract

The *WFS1* gene was first identified in Wolfram Syndrome 1 (WS1), a rare autosomal recessive genetic disorder characterized by severe and progressive neurodegenerative changes. *WFS1*’s role in various cellular mechanisms, particularly in calcium homeostasis and the modulation of endoplasmic reticulum (ER) stress, suggests its potential involvement in the pathogenesis of Alzheimer’s disease (AD) and sleep disorders. Because it is involved in maintaining ER balance, calcium signaling, and stress responses, *WFS1* plays a multifaceted role in neuronal health. Numerous studies have shown that the absence or improper expression of *WFS1* disrupts these cellular processes, leading to neurodegeneration and making neurons more vulnerable. In AD, *WFS1* dysfunction is thought to contribute to the accumulation of amyloid-β (Aβ) plaques and tau tangles, thereby accelerating disease progression. Additionally, *WFS1* plays an essential role in sleep regulation by influencing neuronal excitability and neurotransmitter release, which may explain the sleep disturbances frequently observed in neurodegenerative diseases. Due to its involvement in the pathological mechanisms of AD and sleep disorders, *WFS1* is regarded as a potential early diagnostic marker for these diseases. Further research is required to fully elucidate *WFS1*’s role in the cellular pathway, perhaps facilitating the development of new therapeutic strategies for Alzheimer’s disease and sleep disorders.

## 1. Introduction

The *WFS1* gene, originally identified for its involvement in WS1 [1,2], has since been widely recognized for its broader role in a range of neurological and psychiatric diseases [3,4,5,6]. Indeed, numerous studies have suggested that *WFS1* exerts effects that extend beyond WS1 [5], a rare neurodegenerative disorder characterized by diabetes insipidus DI, diabetes mellitus DM, optic atrophy OA, and deafness D (DIDMOAD), as well as by additional issues such as urinary system abnormalities and disruptions in the endocrine system [7,8,9]. Moreover, WS1 patients often experience a range of severe psychiatric and neurological symptoms, attributed to extensive neurodegenerative processes within the brain [8,10]. Commonly found symptoms include cognitive decline, seizures, anxiety, sleep disorders, suicidal tendencies, and depression [8,11,12,13]. The complexity of *WFS1*’s functions is underscored by its involvement in the regulation of ER homeostasis, particularly in relation to calcium ion signaling and ER stress responses. These cellular processes are critically linked to neurodegeneration, suggesting that *WFS1* could play a pivotal role in maintaining neuronal health. Furthermore, the wolframin protein, encoded by *WFS1*, is implicated in various signaling pathways that influence neuronal excitability and synaptic function. Some studies have suggested a surprising link between alterations in *WFS1* expression and the development of AD, as well as various sleep disorders [4,14,15,16]. Indeed, an overlap in pathophysiological and clinical features between AD and WS1, such as ER stress and calcium dysregulation, has been identified [12,15] (Table 1, Figure 1).

Likewise, several clinical similarities have been observed, although WS1 and AD remain two distinct entities (Table 2 and Table 3, Figure 1).

There are both overlapping and distinct clinical characteristics of WS1 and AD. Shared features are placed in the intersecting area, while unique features specific to each condition are positioned outside the overlap. This visual comparison highlights key clinical manifestations to aid in the differential diagnosis and understanding of these two diseases.

Therefore, a more comprehensive knowledge of the role of *WFS1* in the pathogenesis of AD could not only enhance diagnostic and follow-up strategies for AD but also aid in the development of novel therapeutic targets. Additionally, since sleep disturbances are often early markers of neurodegenerative disorders, exploring the impact of *WFS1* on sleep regulation may provide insights into early diagnostic markers and interventions. Recent studies have suggested that *WFS1* plays a crucial role in regulating sleep patterns [4,14]. Alterations in *WFS1* expression are linked to changes in neuronal excitability and neurotransmitter release, which may contribute to the onset or exacerbation of sleep disorders [4]. Moreover, given that sleep disturbances are common in AD [18] and other neurodegenerative disorders [21], the interaction between *WFS1* dysfunction and sleep regulation offers a promising area for further research. This manuscript aims to explore the current knowledge of *WFS1*’s involvement in AD and sleep disorders, focusing on its potential as a biomarker and therapeutic target.

Through an analysis of existing research and a discussion of recent findings, we seek to clarify the multifaceted roles of *WFS1* in these interconnected fields.

### 1.1. WFS1: Patterns of Expression in Brain Development

In 1998, Strom TM [1] and Inoue H [2] identified the human gene *WFS1* on the short arm of chromosome 4 (chr 4p16). This gene spans 33.4 kilobases (kb) and encodes the transmembrane glycoprotein known as wolframin. *WFS1* consists of eight exons, producing 15 distinct transcripts. Exon 1 is non-coding, while exons 2 through 7 are small coding exons. Exon 8, the largest exon, plays a critical role in wolframin’s functionality as it encodes both the transmembrane region and the terminal carboxyl residue of the *WFS1* protein [1,2]. Wolframin is a hydrophobic transmembrane glycoprotein composed of 890 amino acids (aa) and has a molecular weight of 100 kilodaltons (kDa). It is primarily located in the ER and features nine transmembrane domains, essential for maintaining ER homeostasis. Wolframin is characterized by three distinct structural domains: a central hydrophobic transmembrane domain comprising 9–10 segments and two hydrophilic domains located at the N- and C-termini, situated in the cytoplasm and the lumen of the ER, respectively [1,2,22]. *WFS1* is expressed in various organs and tissues, including the brain, pancreatic cells, heart, lungs, and placenta. Specific brain regions, including the hippocampus, amygdala, brainstem nuclei, and thalamus, exhibit high levels of *WFS1* expression [12,23] (Table 4).

This may likely explain the high prevalence of psychiatric and neurological disorders found in patients with WS1 [13,24]. Wolframin levels are low during the fetal stage, specifically around 14–16 weeks of gestation, and increase until maturity [25,26]. Research has shown that in mice, the expression of *Wfs1* mRNA initiates during the later stages of embryonic development, particularly in regions such as the dorsal striatum and amygdala, which are crucial for regulating emotional responses and motor functions. Postnatally, its expression expands to additional brain regions [23,27]. *WFS1* expression is closely linked to various stages of neuronal differentiation, particularly during myelination. During late childhood to early adolescence, myelination occurs most intensively, although there are significant variations across different brain regions, including the cortex and subcortex [25,26,28]. The observation of high *WFS1* expression levels from early childhood through adolescence indicates that wolframin plays a key role in early brain development, particularly during critical neurodevelopmental periods rather than in later stages of life [12,27] (Figure 2).

These findings are consistent with the understanding that adolescence is characterized by significant processes such as myelination, especially in critical brain areas such as the motor and cingulate cortices, which play a vital role in motor coordination and emotional regulation [3,29].

Furthermore, rapid myelination changes have also been observed during pre-adolescence and adolescence in regions such as the hippocampus, which is crucial for memory formation and learning, as well as in other cortical and subcortical regions involved in cognitive and executive functions. These findings suggest that wolframin may contribute to the proper maturation of neuronal circuits, particularly those undergoing dynamic changes during these stages [27,28,30,31]. Many studies have also found a significant increase in *Wfs1* mRNA expression in the brains of mice, starting from the early stages of life and continuing into early adulthood, while specific areas such as the supraoptic and magnocellular nuclei maintain stable levels postnatally but show a decline thereafter [27,29].

Elevated levels of *Wfs1* mRNA and wolframin have been observed in the amygdala, the Cornu Ammonis 1 (CA1) region of the hippocampus, and other critical areas within the limbic system of rats. These findings suggest that wolframin expression is not only spatially regulated but also varies according to developmental stages and the specific functional demands of different brain regions [27,28,29]. The strong link between *WFS1* expression and neuronal differentiation processes shows how important it is for brain development, especially in the formation and refinement of neural circuits. This is especially evident in regions such as the hippocampus and amygdala, which are essential for memory consolidation, emotional regulation, and stress response [26,27,28,29] (Figure 3).

The expression levels of *WFS1* differ in different brain regions. Regions such as the hippocampus, amygdala, limbic system, and motor cortex show high *WFS1* expression, highlighting its role in memory, emotional regulation, and motor coordination (*Y*-axis: expression level in arbitrary units) [26,27,28,29].

Wolframin’s involvement in the ER stress response may suggest its broader significance in ensuring proper protein folding and cellular homeostasis during periods of intense neurodevelopment. Furthermore, research on severe neurodegenerative disorders, such as WS1, has shown that the absence of wolframin due to *WFS1* mutations leads to neuronal death, particularly in regions such as the cerebellum, optic pathway, and brainstem, despite the widespread expression of *WFS1*. This suggests that certain neuronal populations may be more susceptible to *WFS1* mutations than others, potentially due to their specific functional demands or developmental trajectories [30,32]. In reality, it appears that *WFS1* expression levels do not solely influence neurodegenerative processes, but there are some regional dependencies on other compensatory or protective mechanisms that we still do not fully understand. For example, the expression of *WFS1* is low in the visual system compared to other brain regions, despite optic atrophy being one of the early signs of WS1 (Figure 3). Some parts of the brain, such as the hippocampus and amygdala, may have higher levels of *WFS1* because they depend on the wolframin protein for functions such as calcium homeostasis and ER stress responses. These functions are crucial for maintaining cellular balance and protecting neurons in stressful situations. Contrarily, despite the baseline levels of *WFS1* expression being lower in this area compared to other brain regions, the visual system’s high metabolic demand renders it more susceptible to stress and degeneration in cases of compromised *WFS1* functionality [30,32].

Neurons involved in high metabolic activities, such as those in the hippocampus or cerebellum, could be particularly vulnerable, as *WFS1* is known to play a key role in maintaining ER homeostasis and regulating cellular stress responses. Disruptions in these processes caused by *WFS1* mutations might lead to increased neuronal degeneration or dysfunction in these sensitive areas, contributing to the progression of neurodegenerative diseases [26,33]. Additionally, unidentified proteins or pathways might compensate for the lack of wolframin in some parts of the brain [3,32]. Numerous studies have emphasized the critical role of *WFS1* in neuronal development. Reduced *WFS1* expression has been linked to increased neuronal vulnerability to age-related degeneration, suggesting that *WFS1* plays an evolutionarily conserved role in maintaining neuronal integrity during the aging process [32].

### 1.2. WFS1: Key Regulator of ER Stress and Calcium Homeostasis

Wolframin is specifically located in the membrane of the ER. The ER has a crucial function in ensuring the proper folding and modification of secretory proteins, cell surface receptors, and ER transmembrane proteins [25]. Mutations in *WFS1* cause ER stress because there is an accumulation of misfolded proteins in the ER. An excess of these misfolded proteins activates the unfolded protein response (UPR). The UPR initiates transcriptional and translational processes that aim to restore balance in the ER. However, some physiological (such as the postprandial biosynthesis of insulin) or pathological processes (such as tumors, inflammatory diseases, viral infection, and genetic mutations) create a state of chronic and persistent ER stress, as a result of which the stimulation of UPR causes cellular apoptosis [7,25,33]. Thus, excessive ER stress can lead to the death of pancreatic cells and the characteristic neuronal modification that are hallmark features of WS1 [34]. The UPR is a complex mechanism that activates three transmembrane proteins in the ER, which function as stress sensors: inositol-requiring enzyme 1 (IRE1), RNA-dependent protein kinase-like endoplasmic reticulum kinase (PERK), and activating transcription factor 6 (ATF6). These transducers have dual roles: promoting cellular adaptation and initiating apoptosis [32]. Furthermore, some studies have shown that an excess of UPR activity stimulates the synthesis of an immunoglobulin-binding protein (BIP) that aids the correct folding of accumulated proteins. Under normal conditions, BIP and other ER chaperones keep their luminal domains in an inactive state. When there is a high level of UPR activity in the ER, BIP is synthesized to assist in the proper folding of accumulated proteins [35,36]. During physiological stress, the oligomerization and autophosphorylation of IRE1 occur, followed by the splicing of X-box binding protein 1 messenger RNA (XBP-1 mRNA). This process produces a transcription-ready form of mRNA, known as sXBP-1. When sXBP-1 activates, it produces X-box binding protein 1 (XBP-1), a transcription factor that, upon reaching the nucleus, initiates the expression of specific genes involved in ER-associated degradation (ERAD). Thus, BIP activates cytoprotective mechanisms by restoring protein homeostasis [33,34,35,36]. The excessive activation of IRE1 under pathological conditions triggers apoptosis pathways by recruiting TNF receptor-associated factor 2 (TRAF2) and phosphorylating apoptosis signal-regulating kinase 1 (ASK1). ASK1, in turn, phosphorylates c-Jun N-terminal kinase (JNK), which subsequently promotes apoptosis [32,33,34,35,36]. A key transmembrane protein called PERK plays a crucial role in ER stress and starts the phosphorylation of eukaryotic initiation factor 2 alpha (eIF2α), which is a translation initiation factor in eukaryotes. In this phosphorylation event, the biosynthetic activity of the ER decreases, while translation of Activating Transcription Factor 4 (ATF4) and mRNAs encoding apoptosis-antagonizing transcription factors (AATF) increases. ATF4 activates genes essential for amino acid transport and metabolism, glutathione production, and antioxidant responses. Pathological ER stress activates the ATF4-activating transcription factor 3 (ATF3)–C/EBP Homologous Protein (CHOP) complex, resulting in apoptosis. In contrast, the AATF factor promotes cell survival [23,25,36]. Activating transcription factor 6 (ATF6) plays a critical role in regulating the UPR. During ER stress, the protein BIP dissociates, allowing ATF6 to translocate to the Golgi apparatus. In the Golgi, specific proteases cleave ATF6, generating an active transcription factor within the cytoplasm. When ATF6 is turned on, it goes to the nucleus to improve protein folding, processing, and degradation by increasing transcription factors that maintain ER homeostasis [32,36]. *WFS1* acts as a negative regulator of the UPR under normal ER stress conditions. It inhibits ATF6, lowers the activation of the ER stress response element (ERSE), and keeps E3 ubiquitin ligase HMG-CoA reductase degradation protein 1 (HRD1) stable. HRD1 is a key protein that helps break down 3-hydroxy-3-methylglutaryl-Coenzyme A reductase (HMG-CoA reductase). As a result, *WFS1* helps to suppress stress signals. In WS1, on the other hand, too much activation of ATF6 causes more expression of genes related to apoptosis, such as CHOP, ATF4, BIP, and sXBP1, while insulin-related genes are expressed less [32,34]. Wolframin deficiency is thought to lead to an increased UPR, exacerbating neurodegenerative damage through the activation of ER stress mechanisms [25]. Therefore, wolframin may be considered a factor crucial for the survival of neurons and overall brain health (Table 5).

The role of *WFS1* in calcium regulation and the effects of its dysfunction include impacts on ER mitochondrial signaling, calcium homeostasis, and autophagy. The dysfunction of *WFS1* is linked to conditions such as WS1 and neurodegenerative diseases [12,22,23,25,28].

### 1.3. WFS1 Dysfunction: A Link Between WS1, AD, and Sleep Disorders

AD is the most common cause of dementia worldwide, affecting over 55 million people, a number that continues to rise due to the aging global population. According to the World Alzheimer Report 2021, the prevalence of AD is expected to increase dramatically in the coming decades [37]. AD is characterized by a progressive decline in cognitive abilities, including memory loss, impaired communication, changes in reasoning and judgment, and often behavioral disturbances [37,38]. Pathologically, AD is defined by the accumulation of Aβ plaques and neurofibrillary tangles composed of hyperphosphorylated tau protein, leading to widespread neuronal loss, synaptic dysfunction, and brain atrophy. These pathological structures trigger neuroinflammatory responses and exacerbate oxidative stress, contributing to the neurodegeneration typical of AD [39]. In parallel, WS1, a rare genetic neurodegenerative disorder, shares molecular mechanisms with AD, particularly the disruption of ER homeostasis and calcium signaling. While WS1 is primarily characterized by diabetes insipidus, diabetes mellitus, optic atrophy, and deafness, it also involves significant neuronal degeneration. Interestingly, the same *WFS1* dysfunction observed in WS1 has been implicated in the pathogenesis of AD [5,40]. One of the central molecular features of both AD and WS1 is the disruption of calcium homeostasis and the induction of ER stress. In AD, Aβ and tau pathology contribute to neuronal dysfunction by exacerbating these processes, leading to synaptic failure and neuronal apoptosis. *WFS1* plays a crucial role in mitigating these effects by maintaining ER homeostasis and regulating calcium dynamics. Mutations or altered expression of *WFS1* may worsen ER stress and disrupt calcium signaling, accelerating the accumulation of Aβ plaques and tau tangles, which are the hallmark features of AD [41,42]. Recent studies have highlighted the potential overlap between *WFS1* dysfunction in WS1 and AD. The *WFS1* gene, initially known for its role in regulating cellular stress responses in WS1, is now recognized for its broader implications in neurodegeneration, particularly in AD. The ER stress responses and calcium dysregulation seen in both conditions suggest that investigating the role of *WFS1* in AD could lead to novel therapeutic strategies aimed at reducing neurodegeneration and improving cognitive outcomes [5]. The key molecular pathways altered by *WFS1* dysfunction are summarized in Table 6.

### 1.4. ER Stress and the Unfolded Protein Response Alterations

Wolframin plays a crucial role in AD by modulating ER stress. Mutations in *WFS1* result in chronic ER stress and an altered UPR in both AD and WS1 [41,42]. In AD, the accumulation of Aβ and tau proteins not only disrupts synaptic signaling but also triggers ER stress, thereby intensifying the UPR. Persistent UPR activation shifts toward a pro-apoptotic state, contributing significantly to neuronal death. Consequently, deficiencies or dysfunctions in wolframin increase neuronal susceptibility to apoptosis. This ongoing ER stress exacerbates neurodegeneration and plays a significant role in the progressive cognitive decline observed in AD patients [43,44].

### 1.5. Calcium Homeostasis and Synaptic Dysfunction

Alterations in calcium homeostasis significantly influence the onset and progression of AD. Calcium ions (Ca^2+^) play a key role in neuronal excitability, neurotransmitter release, and synaptic plasticity, all of which are closely linked to cognitive processes such as learning and memory [45]. In AD, disruptions in calcium signaling lead to excessive excitotoxicity, synaptic failure, and neuronal death [45,46]. Recent studies have suggested that disruptions in calcium homeostasis are not just a result of Aβ and tau pathology but may also act as a contributing factor that triggers these molecular changes [47]. These severe changes are closely associated with Aβ plaques and tau pathology. Aβ plaques, in fact, create pores in neuronal membranes, leading to abnormal calcium influx. Meanwhile, tau pathology interferes with calcium-binding proteins [47,48]. The excessive calcium influx into neurons triggers several deleterious processes, including overactivation of calcium-dependent enzymes such as proteases, phospholipases, and endonucleases, leading to mitochondrial damage, increased production of reactive oxygen species (ROS), and ultimately, neuronal apoptosis [49]. Wolframin plays a protective role by mitigating these effects through two key mechanisms: maintaining appropriate Ca^2+^ levels in the ER and coordinating calcium signaling between the ER and mitochondria, both essential for ATP production and cell survival [25,50]. This delicate calcium homeostasis is disrupted when wolframin is absent or poorly expressed due to *WFS1* mutations, as seen in WS1 and possibly in AD. Inevitably, this dysregulation leads to severe mitochondrial dysfunction, increased production of ROS, and neuronal damage. The impaired Ca^2+^ buffering caused by the absence or reduced expression of wolframin further exacerbates the release of cytotoxic factors, marking the onset of neurodegeneration and cognitive decline [5,12].

### 1.6. Mitochondrial Dysfunction and Dysregulated ER–Mitochondria Communication

Wolframin also contributes to the pathogenesis of AD through the regulation of mitochondrial function and the exchanges between the ER and mitochondria. ATP produced by mitochondria is essential for synaptic activity and cellular metabolism [1,2,22,50]. The progression of AD is closely related to mitochondrial dysfunction, which contributes to oxidative stress, impaired energy metabolism, and neuronal death [51]. Wolframin plays a crucial role in mitochondria-associated membranes (MAMs), essential for the transfer of Ca^2+^ from the ER to the mitochondria, influencing cellular bioenergetics and supporting neuronal survival. In AD, communication between the ER and mitochondria is severely disrupted, leading to impaired calcium transfer [51]. *WFS1* mutations destabilize MAMs, reducing Ca^2+^ uptake by mitochondria. This leads to mitochondrial depolarization, impaired ATP production, and increased generation of ROS [49,50,51]. As AD progresses, the accumulation of ROS causes oxidative damage to mitochondrial DNA and proteins, exacerbating neurodegeneration. The resulting mitochondrial dysfunction further compromises neuronal survival, accelerating cognitive decline in AD [49,50,51]. Additionally, mitochondrial dynamics are severely impaired, leading to dysregulated fission, fusion, and mitophagy. This exacerbates mitochondrial dysfunction by promoting the accumulation of damaged organelles, which cannot be properly eliminated, leading to further oxidative damage and energy deficits. Prolonged ER stress, commonly observed in AD, further disrupts the calcium signaling between the ER and mitochondria by activating the UPR. When chronically activated, the UPR response disrupts ER homeostasis and contributes to neurodegeneration [5,12].

### 1.7. Neuroinflammation and Oxidative Stress

AD is characterized by neuroinflammation, where there is activation of microglial cells and astrocytes secondary to Aβ plaques and tau tangles. The inflammatory response is intended to promote the elimination of toxic protein aggregates. However, when prolonged, it leads to the release of pro-inflammatory cytokines, causing further neuronal damage). There is a close link between oxidative stress and neuroinflammation, as the ROS generated during oxidative stress further activate microglia and astrocytes, thereby amplifying the inflammatory response [51,52,53,54]. Conversely, chronic neuroinflammation further alters mitochondrial function, increasing the production of ROS [52,53,54]. Neuronal damage thus progresses, facilitating the advancement of AD. In this vicious cycle, wolframin plays an important role, as its absence or reduced expression exacerbates oxidative stress and, consequently, mitochondrial dysfunction. This results in an increase in ROS production, leading to oxidative damage to proteins, lipids, and DNA, thereby accelerating the neurodegenerative process in AD [5]. Wolframin plays a key role in the neuroinflammatory processes of AD by regulating ER stress and responses to oxidative stress [42]. On the other hand, mitochondrial dysfunction caused by *WFS1* mutations also contributes to the activation of neuroinflammatory pathways, as it releases damage-associated molecular patterns (DAMPs), which trigger immune responses in the brain [55]. In this context, wolframin could help mitigate the neuroinflammatory response [5,12]. These data suggest that restoring *WFS1* function or modulating its pathways could be beneficial in combating both oxidative stress and inflammation.

### 1.8. Dysregulation of Autophagy and Protein Clearance

In AD, autophagy, a process heavily regulated by *WFS1*, plays a crucial role, as its dysfunction is one of the pathogenic mechanisms underlying the disease. The absence or reduced expression of wolframin results in the cell’s inability to effectively eliminate Aβ plaques and tau tangles, leading to their accumulation in the brain, exacerbation of neuronal stress, and synaptic failure [5]. Autophagy is a tightly regulated process, controlled by key signaling pathways such as the mechanistic target of rapamycin (mTOR) pathway, which senses nutrient availability and cellular energy levels. Dysregulation of mTOR signaling due to *WFS1* dysfunction further compromises the autophagic process, exacerbating neurodegeneration. Furthermore, the accumulation of damaged mitochondria caused by wolframin dysfunction, combined with the reduced clearance of protein aggregates, creates a toxic environment that accelerates neuronal death and contributes to cognitive decline in AD [56,57,58,59].

### 1.9. Altered Aβ Production and WFS1

Recent studies have suggested a direct link between *WFS1* and the production of Aβ, one of the main pathological features of AD. Aβ derives from the sequential cleavage of the amyloid precursor protein (APP) by beta-site amyloid precursor protein cleaving enzyme 1 (BACE1) and γ-secretase [60,61]. Several studies have shown that *WFS1* influences this process, as it regulates the function of presenilins, which are the catalytic subunits of γ-secretase [60,61]. Mutations in the presenilin genes (PSEN1 and PSEN2) have been identified in familial forms of AD, leading to altered γ-secretase function [62]. Presenilins also contribute to calcium dysregulation and ER stress, processes influenced by *WFS1* [63]. The deficiency of *WFS1* leads to an increase in the production of Aβ, likely due to the dysregulation of presenilin-dependent mechanisms [64]. In the case of *WFS1* mutations, the increased activity of γ-secretase leads to excessive production of Aβ peptides, resulting in plaque formation and acceleration of AD progression. This mechanism is further complicated by disruptions in Ca^2+^ homeostasis, which promote excessive cleavage of APP [64,65]. Thus, a vicious cycle is established between the overproduction of Aβ, increased neuronal damage, and further stress on cellular homeostasis. Moreover, an exaggerated UPR triggered by *WFS1* dysfunction exacerbates ER stress, leading to additional accumulation of misfolded proteins and, consequently, Aβ aggregation. Disturbed autophagy, calcium dysregulation, and increased production of Aβ play, thus, a key role in the pathogenesis of AD [12,64,65].

### 1.10. WFS1, Tau Phosphorylation, and Neurofibrillary Tangles

Neurofibrillary tangles, composed of hyperphosphorylated tau protein, are a typical marker of AD. Tau is a microtubule-associated protein that stabilizes neuronal cytoskeletal structures. In AD, tau undergoes hyperphosphorylation, leading to the formation of insoluble tangles that interfere with normal neuronal function and ultimately contribute to cell death [57,66]. It is believed that wolframin may influence the phosphorylation of tau, as it plays an important role in ER homeostasis and the prevention of oxidative stress [12]. Chronic ER stress and oxidative damage alter the activity of kinases such as glycogen synthase kinase-3β (GSK-3β), which phosphorylates tau [67]. In the case of *WFS1* deficiency, there is an exacerbation of stress conditions, leading to an increase in tau hyperphosphorylation and the formation of tangles. When tau tangles accumulate in areas of the brain, such as the hippocampus and the cortex, severe synaptic dysfunction and memory deterioration occur, both of which are characteristic of AD [5,12].

### 1.11. Altered Synaptic Plasticity and Cognitive Decline

The formation of memory and cognitive function are closely related to synaptic plasticity, which represents the ability of synapses to strengthen or weaken in response to activity. In AD, synaptic plasticity is severely impaired due to the toxic effects of Aβ and tau proteins. These proteins disrupt synaptic signaling by compromising receptor trafficking, destabilizing the cytoskeleton, and inducing oxidative stress, ultimately undermining synaptic integrity [68]. A further deterioration occurs due to alterations in calcium signaling, which contribute to the breakdown of processes such as long-term potentiation (LTP), which is essential for learning and memory [69]. Wolframin plays an essential role in maintaining synaptic plasticity and the stability of neuronal networks through its regulation of calcium homeostasis. The inability of neurons to communicate with each other and synaptic dysfunction caused by the altered expression of wolframin represent key factors in the memory loss and cognitive decline observed in patients with AD [68].

### 1.12. The Role of WFS1 in Sleep Regulation: Mechanisms and Implications for Neurodegenerative and Psychiatric Disorders

Recent research has shown that *WFS1* plays a key role in regulating sleep and circadian rhythms due to its involvement in neuronal activity and calcium homeostasis [69].

#### Dopaminergic Neurons and Sleep Regulation

Hao et al. evaluated the effects of *WFS1* expression deficiency on dopamine 2-like receptor neurons (Dop2R), which play a pivotal role in promoting alertness. Altered expression of *WFS1* leads to reduced sleep and disrupted circadian rhythms. Specifically, *WFS1* deficiency increases the excitability of these neurons, resulting in increased wakefulness and shortened sleep duration. Thus, *WFS1* regulates sleep through dopaminergic signaling, a key pathway in sleep–wake homeostasis [70]. Recent studies further confirm that inhibition of dopamine synthesis can partially correct the sleep deficits caused by *WFS1* loss, highlighting the importance of dopamine in modulating sleep–wake cycles [70,71]. This finding underscores the central role of dopamine in the regulation of both sleep and alertness, with *WFS1* acting as a modulator of this system [70]. A deficiency or low expression of *WFS1* also contributes to sleep problems by increasing the excitability of neurons, which can be attributed to *WFS1*’s role in maintaining calcium homeostasis, particularly in Dop2R neurons. *WFS1* regulates intracellular calcium levels in the ER, and its absence disrupts calcium balance, leading to hyperexcitability and impaired sleep [70,71]. Moreover, *WFS1* mutations have been linked to significant disruptions in synaptic plasticity, the mechanism that underpins learning and memory. Synaptic plasticity is essential for the adaptability of neural circuits, and its disruption by *WFS1* deficiency could account for the sleep disturbances and cognitive impairments found in patients with WS1. Impaired synaptic plasticity may exacerbate sleep-related issues by reducing the brain’s capacity to adapt to sleep–wake signals, leading to an even more pronounced dysregulation of sleep patterns in affected individuals. This interaction between *WFS1*, dopamine signaling, calcium regulation, and synaptic plasticity highlights the multifaceted role of *WFS1* in maintaining neuronal and sleep health. Consequently, the sleep disturbances observed in patients with WS1 likely arise from the combined effects of disrupted dopaminergic signaling, calcium dysregulation, and impaired synaptic plasticity [72,73].

### 1.13. Calcium Homeostasis and ER Stress in Sleep Regulation

A lot of research has shown that when *WFS1* is missing or not expressed properly, it seriously affects calcium homeostasis, which in turn affects how neurons work. This calcium imbalance is one of the main causes of sleep disorders observed in people with mutations in the *WFS1* gene. In particular, receptors such as the ryanodine receptor (RyR) and the inositol 1,4,5-trisphosphate receptor (Itpr), which control the release of calcium from the ER, are needed to get back to a normal sleep pattern. Experimental models have observed that modifying these receptors to restore calcium balance resolves sleep deficits induced by the lack of *WFS1* [74,75].

#### 1.13.1. WFS1 and Sleep Apnea

*WFS1* expression affects not only circadian rhythms but also plays a significant role in the development of sleep apnea. New research indicates that patients with *WFS1* mutations have a higher likelihood of developing both obstructive sleep apnea (OSA) and central sleep apnea (CSA). This is because *WFS1* is highly expressed in the brainstem, a region critical for the automatic regulation of breathing during sleep. Recent studies have highlighted that *WFS1* interacts with essential mechanisms involved in the regulation of ventilation during sleep [76,77]. It has been hypothesized that *WFS1*’s role in sleep apnea extends beyond mere respiratory control. *WFS1* mutations may influence the sensitivity of chemoreceptors and the brainstem’s response to fluctuations in oxygen and carbon dioxide levels during sleep, potentially leading to irregular respiratory patterns. Moreover, *WFS1*’s involvement in maintaining calcium homeostasis and ER stress might further impact the stability of these vital respiratory functions [76,77].

#### 1.13.2. WFS1 in Sleep Disorders

Sleep disorders caused by mutations in the *WFS1* gene are often associated with psychiatric conditions, such as anxiety, depression, bipolar disorder, and suicidal tendencies. Furthermore, heterozygous carriers of *WFS1* mutations exhibit a higher prevalence of psychiatric disorders compared to the general population. Research has shown that heterozygous subjects with *WFS1* mutations have a 26-fold increased risk of hospitalization for psychiatric disorders, many of which are linked to sleep disturbances [4,5]. The influence of *WFS1* on calcium homeostasis and the regulation of the ER is also crucial to the proper functioning of the hypothalamic–pituitary–adrenal axis (HPA), which is responsible for regulating the body’s response to stress. When *WFS1* mutations disrupt this axis, the resulting dysregulation exacerbates both sleep disorders and stress responses. This creates a vicious cycle where poor sleep leads to increased stress, which in turn worsens sleep disturbances and psychiatric symptoms. *WFS1* mutations may also impair the secretion of cortisol, a key hormone in the HPA axis that helps manage stress. Dysregulation of cortisol levels further disrupts circadian rhythms and contributes to the onset or worsening of psychiatric conditions [5,12,76] (Table 7).

The psychiatric disorders found in WS1 and AD are summarized in Figure 4.

The above shows the percentage of neurological and psychiatric disorders between WS1 and AD patients. Cognitive decline is reported in 100% of AD patients, while anxiety and depression are more frequent in WS1 patients. Sleep disturbances affect both groups similarly, while seizures and psychosis are less common in WS1 patients.

## 2. Conclusions

Numerous studies have shown the key role that the *WFS1* gene plays in both WS1 and certain neurodegenerative and psychiatric disorders, particularly AD. While WS1 and AD are distinct diseases, they share important molecular mechanisms, including disruptions in ER homeostasis, alterations in calcium regulation, and increased ER stress, all of which contribute to neurodegeneration in both conditions. In both cases, the absence or reduced expression of *WFS1* leads to severe neuronal and psychiatric damage. Nevertheless, it is crucial to ascertain whether these shared mechanisms are primary disease drivers or convergent cellular responses specific to each disorder. If the mechanisms behind WS1 and AD are mechanistically similar, WS1 models could help AD research, and vice versa. However, unique cellular alterations in each disease might require condition-specific therapeutic adaptations.

Therapies that target calcium dysregulation and ER stress may alleviate neurodegenerative symptoms in both conditions. However, the primary triggers differ between AD and WS1, with Aβ plaques and tau tangles in AD and *WFS1* mutations in WS1, which means therapies designed for one disorder might not fully benefit the other.

Several factors implicated in WS1 pathogenesis, such as calcium mishandling, ER stress, and UPR activation, are also present in other degenerative diseases. However, these shared features likely represent common end-stage responses rather than primary disease drivers. WS1, specifically, is characterized by a distinctive *WFS1* mutation, leading to early and chronic ER stress, which is not a universal feature in all degenerative diseases. While WS1 models offer valuable insights into cellular stress responses, their pathology and progression differ substantially from other disorders, precluding their direct applicability to all neurodegenerative conditions.

In WS1, mutations in the *WFS1* gene cause chronic ER stress and calcium homeostasis dysregulation, resulting in neurodegeneration. Similar alterations in these pathways, exacerbated by Aβ plaques and tau tangles, characterize the progressive neurodegeneration typical of AD. Thus, *WFS1* is believed to play a key role in modulating cellular stress responses in both diseases.

In AD, altered *WFS1* expression and function have been observed, particularly in brain regions such as the hippocampus, which is vulnerable to Aβ and tau pathology. These findings suggest that *WFS1* deficiency may contribute to AD pathogenesis by intensifying ER stress and calcium dysregulation. However, further research is needed to clarify whether WFS1 deficiency is a causative factor or a secondary consequence of AD-related stressors.

One of the primary therapeutic objectives for both WS1 and AD is restoring calcium homeostasis. In both conditions, the dysregulation of calcium signaling is critical to neurodegenerative processes. Targeting calcium regulation, for example, by modulating receptors such as the RyR and the Itpr, has shown promise in experimental models. Restoring proper calcium dynamics could help alleviate both sleep disturbances and cognitive impairments, particularly by improving synaptic plasticity and reducing neuronal excitability.

Reducing ER stress is another crucial therapeutic strategy. Both WS1 and AD are characterized by excessive UPR activation due to misfolded proteins in the ER. Neurodegeneration might be slowed down by using treatments that focus on maintaining ER homeostasis and blocking pathways that cause cells to die (for example, the ATF4–ATF3–CHOP complex). Research has shown that mitigating chronic ER stress may prevent the progression of both diseases and protect vulnerable neuronal populations.

*WFS1* has therefore emerged as a promising therapeutic target for both AD and WS1. Therapies that either increase *WFS1* activity or compensate for its deficiency could mitigate cellular stress, restore calcium balance, and prevent progressive neuronal injury. Such approaches hold promise as they address shared molecular pathways, while also allowing for disorder-specific adjustments to optimize therapeutic outcomes.

Moreover, addressing dopaminergic signaling dysregulation is another critical therapeutic goal, particularly for the sleep disturbances associated with both WS1 and AD. WFS1 is involved in the regulation of dopaminergic neurons, and its deficiency leads to heightened neuronal excitability and disrupts circadian rhythms. Therapies aimed at modulating dopamine levels or receptor activity could help correct sleep disturbances, thereby improving overall neuronal function and potentially slowing neurodegenerative processes.

Experimental models have demonstrated that modulating receptors such as the RyR and Itpr can enhance synaptic plasticity and decrease neuronal excitability, potentially alleviating sleep and cognitive issues.

Lastly, future therapies could focus on balancing the HPA axis, particularly in patients with psychiatric comorbidities. WFS1 mutations impact the HPA axis, exacerbating stress responses and further disrupting sleep patterns. Stabilizing cortisol levels and restoring proper circadian regulation could break the cycle of stress-induced sleep disruption, which often worsens psychiatric symptoms in these patients.

In summary, the development of therapeutic strategies for AD and WS1 must focus on regulating the dopaminergic and HPA axes, reducing ER stress, and maintaining calcium homeostasis. These approaches aim not only to slow neurodegeneration but also to improve quality of life by addressing sleep disturbances and psychiatric symptoms often linked to these conditions. Further studies are required to refine these therapeutic targets and to develop effective treatments.

## Figures and Tables

**Figure 1 biomolecules-14-01389-f001:**
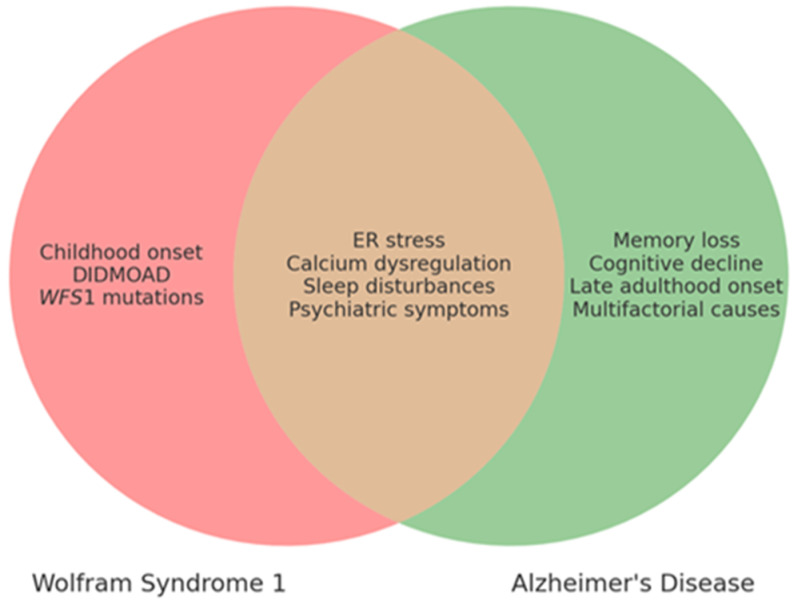
Comparison of clinical features of WS1 and AD.

**Figure 2 biomolecules-14-01389-f002:**
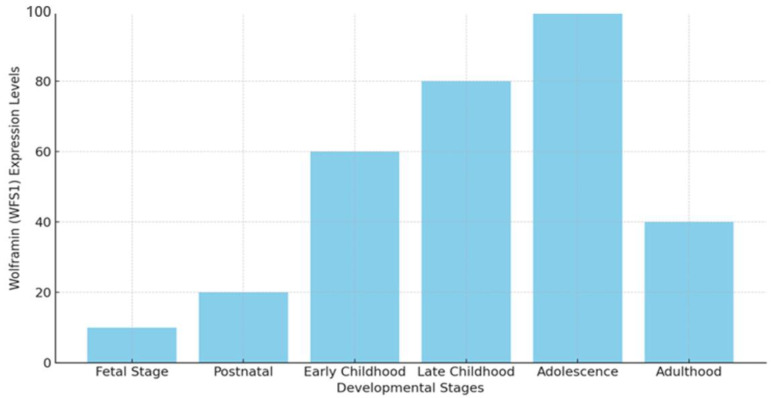
Expression of *WFS1* at various developmental stages. Expression levels of *WFS1* differ across different developmental stages, from the fetal stage to adulthood, highlighting its peak during adolescence and its decline in adulthood (*Y*-axis: expression level in arbitrary units). The data represent *WFS1* expression in the hippocampus and thalamus regions [12,23,25,26,27,28].

**Figure 3 biomolecules-14-01389-f003:**
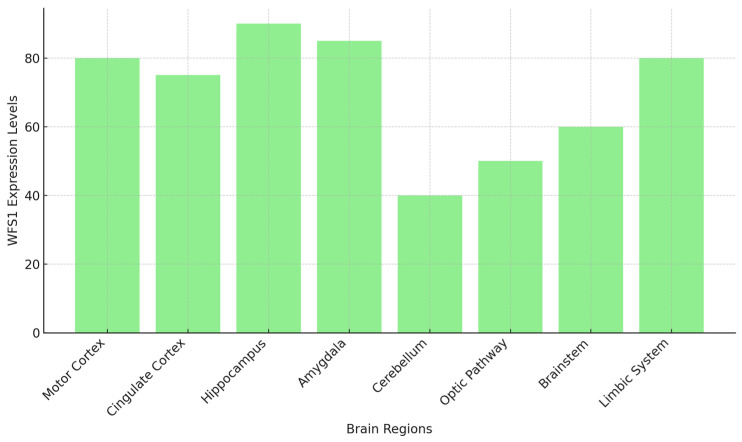
The expression levels of *WFS1* in different brain regions. Regions such as the hippocampus, amygdala, limbic system and motor cortex show high *WFS1* expression, highlighting its role in memory, emotional regulation, and motor coordination (*Y*-axis: Expression Level in arbitrary units) [26,27,28,29].

**Figure 4 biomolecules-14-01389-f004:**
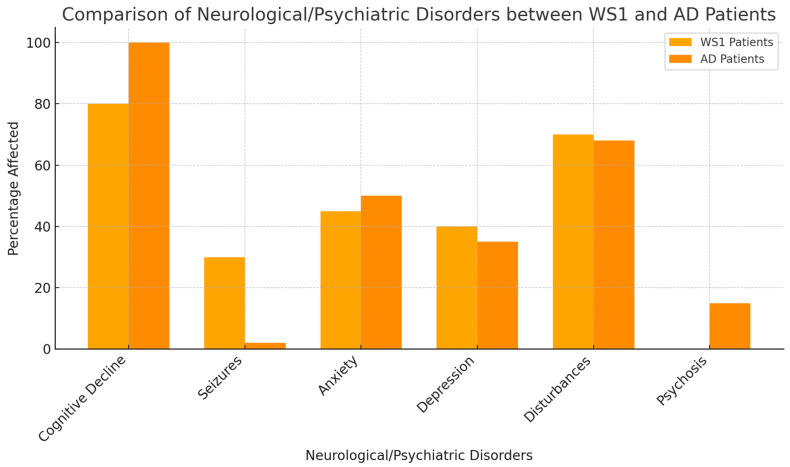
Frequency of neurological and psychiatric disorders in WS1 and AD patients.

**Table 1 biomolecules-14-01389-t001:** Comparison of Pathological Mechanisms Shared Between WS1 and AD.

Mechanism	WS1	AD
ER stress	Chronic ER stress, apoptosis	Excessive UPR, neuronal death
Calcium dysregulation	Dysregulated calcium signaling	Calcium influx, excitotoxicity
Neurodegeneration	Neuronal death in the brainstem and cerebellum	Synaptic dysfunction, brain atrophy
Aβ and Tau accumulation	Absent	Present (Aβ plaques, tau tangles)

Overview of shared and distinct pathological mechanisms between WS1 and AD, highlighting similarities such as ER stress, calcium dysregulation, and neurodegeneration, as well as differences such as the absence of Aβ and Tau accumulation in WS1 [12].

**Table 2 biomolecules-14-01389-t002:** Clinical Characteristics of WS1 and AD.

Clinical Feature	WS1	AD
Age of Onset	Childhood to adolescence	Late adulthood (65+)
Main Symptoms	DIDMOAD, neuropsychiatric disorders, urinary/endocrine issues	Memory loss, cognitive decline, behavioral changes
Neurodegeneration	Optic pathway, cerebellum, brainstem alterations	Hippocampus, cortex
ER Stress Involvement	Yes	Yes
Calcium Dysregulation	Yes	Yes
Sleep Disturbances	Common (up to 70%)	Common
Psychiatric Symptoms	Anxiety, depression, seizures	Depression, anxiety, agitation
Genetic Cause	*WFS1* gene mutation	Multifactorial (genetic/environmental)

Clinical differences between WS1 and AD, such as age of onset and main symptoms, along with shared features such as neurodegeneration, ER stress, and psychiatric symptoms [5,7,12].

**Table 3 biomolecules-14-01389-t003:** Frequency of Neurological and Psychiatric Disorders in WS1 and AD Patients.

Neurological/Psychiatric Disorders	Percentage of WS1 Patients Affected	References (WS1)	Percentage of AD Patients Affected	References (AD)
Cognitive Decline	60%	Barrett et al. [8]; Rohayem et al., 2011 [9]	100%	Huang et al. [17]
Seizures	30%	Chaussenot et al. [10]	Rare	Prodhan AHMSU et al. [18]
Anxiety	50%	Barrett et al. [8]	40–60%	Cacabelos et al. [16]
Depression	40%	Scolding et al. [12]	40–50%	Huang et al. [17]
Sleep Disturbances	70%	Munshani et al. [13]	60–70%	Prodhan AHMSU [18]
Psychosis	20%	Swift et al. [19]	10–15%	Muhlbauer et al. [20]

Comparison of the occurrence of neurological and psychiatric disorders in WS1 and AD patients, illustrating the differing prevalence of these conditions in the two groups [5,7,12].

**Table 4 biomolecules-14-01389-t004:** *WFS1* Expression in Brain Regions.

Brain Region	Level of WFS1 Expression
Hippocampus	High
Amygdala	High
Brainstem	Moderate
Thalamus	High
Cerebellum	Low
Prefrontal cortex	Moderate

Levels of *WFS1* gene expression in different brain regions, with high expression in areas such as the hippocampus and amygdala [12,23].

**Table 5 biomolecules-14-01389-t005:** Role of *WFS1* in Calcium Regulation.

Function of *WFS1*	Effect of Dysfunction
Regulates calcium in the ER	Calcium imbalance, hyperexcitability, impaired protein folding
Coordinates ER mitochondria calcium signaling	Disrupted ATP production, mitochondrial damage, oxidative stress
Modulates calcium influx from the extracellular space	Altered calcium signaling, impaired synaptic transmission
Maintains synaptic plasticity	Synaptic failure, cognitive decline, neurodegeneration
Regulates stress response pathways (e.g., UPR)	Chronic ER stress, increased apoptosis, neuronal dysfunction
Supports autophagy and protein clearance	Disrupted autophagy, protein aggregation, neurodegenerative diseases

**Table 6 biomolecules-14-01389-t006:** Key Molecular Pathways Altered by *WFS1* Dysfunction.

Molecular Pathway	Normal Role of *WFS1*	Effect of *WFS1* Dysfunction
Unfolded Protein Response	Maintains ER homeostasis, regulates UPR sensors (ATF6, IRE1, PERK)	Chronic ER stress, prolonged UPR, increased apoptosis
Calcium Homeostasis	Controls ER calcium levels and signaling between ER and mitochondria	Calcium dysregulation, excitability, mitochondrial dysfunction
Mitochondrial Function	Supports ER–mitochondria communication for ATP production	Impaired ATP production, oxidative stress, ROS increase
Autophagy and Protein Clearance	Prevents protein aggregation via proper folding and degradation	Misfolded proteins, amyloid-β plaques, tau tangles
Dopaminergic Signaling	Modulates neurotransmitter release and neuronal excitability	Altered release, excitability, sleep disturbances
Neuroinflammation	Mitigates stress-induced inflammation	Activation of inflammatory pathways, cytokine release

The table outlines the normal roles of *WFS1* across various molecular pathways and the effects of its dysfunction, emphasizing the pathological changes found in conditions related to *WFS1* impairment [12,22,23,25,28].

**Table 7 biomolecules-14-01389-t007:** *WFS1* and Sleep Disorders.

Sleep Disorder	*WFS1* Dysfunction Mechanism
Sleep apnea	Disrupted brainstem regulation of breathing
Insomnia	Hyperexcitability due to calcium dysregulation
Circadian rhythm disorders	Altered dopaminergic signaling
REM sleep behavior disorder	Impaired neuronal excitability

The links between *WFS1* dysfunction and various sleep disorders, including sleep apnea and insomnia [5,12,76].

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
