# Peer review of "Beyond Wolfram Syndrome 1: The WFS1 Gene’s Role in Alzheimer’s Disease and Sleep Disorders"

_biomolecules, 2024, doi:10.3390/biom14111389_

Round 1

Reviewer 1 Report

Comments and Suggestions for Authors

This is a review about similarities of Wolfram syndrome and Alzheimer disease.  The topic is relevant and the manuscript has systematically reviewed the available relevant literature. I have only small remarks that,  in my opinion, would improve this manuscript.

On figure 2, what are the units on Y-axis? And, as WFS1 expression is very heterogeneous, in what tissue or region was WFS1 expression data copmpared?

Same goes to Figure 3, what units are on Y-axis? The visual system is one of the first that is detoriated in WS patients, however, the level of WFS1 expression is rather one of the lowest in this region according to this image. Why is WFS1 expression level not correlated with severity of neuronal degeneration in a given region? This should be discussed in the text.

And in more general terms- the manuscript would benefit by including a stronger conclusion and future perspective. How similar AD and WS are at the end? It is clear from this review that AD and WS share a common pathology - calcium mishandling and ERstress/UPR. Is this cellular pathology the underlying mechanism of these disorders? Or are there a different underlying cellular alterations in WS and AD that in turn lead to similar changes in calcium mishandling and ER stress?  Answer to this question is important, as it can suggest whether models of WS can be applied for AD related research and vice versa.   And consequently- can a treatment effective in WS patients be suggested for AD patients? Conversely, can AD drug/treatment be effective in WS patients? If the disorders are so similar, the treatments should be effective for both disorders. Concluding remarks should address these questions more clearly. 

Calcium mishandling, ER stress and UPR are probably universal feature of any degenerative disease. At one stage or the other, cells most likely display calcium mishandling before their death, regardless of underlying primary pathological trigger.  Are all degenerative disorders similar to WS?

I suggest a publication of this manuscript after very minor changes, mostly in concluding remarks.

Author Response

REFEREE N. 1

This is a review about similarities of Wolfram syndrome and Alzheimer disease.  The topic is relevant and the manuscript has systematically reviewed the available relevant literature. I have only small remarks that,  in my opinion, would improve this manuscript.

On figure 2, what are the units on Y-axis? And, as WFS1 expression is very heterogeneous, in what tissue or region was WFS1 expression data compared?

Same goes to Figure 3, what units are on Y-axis? The visual system is one of the first that is detoriated in WS patients, however, the level of WFS1 expression is rather one of the lowest in this region according to this image. Why is WFS1 expression level not correlated with severity of neuronal degeneration in a given region? This should be discussed in the text.

And in more general terms- the manuscript would benefit by including a stronger conclusion and future perspective. How similar AD and WS are at the end? It is clear from this review that AD and WS share a common pathology - calcium mishandling and ERstress/UPR. Is this cellular pathology the underlying mechanism of these disorders? Or are there a different underlying cellular alterations in WS and AD that in turn lead to similar changes in calcium mishandling and ER stress?  Answer to this question is important, as it can suggest whether models of WS1 can be applied for AD related research and vice versa.   And consequently- can a treatment effective in WS1 patients be suggested for AD patients? Conversely, can AD drug/treatment be effective in WS1 patients? If the disorders are so similar, the treatments should be effective for both disorders. Concluding remarks should address these questions more clearly. 

Calcium mishandling, ER stress and UPR are probably universal feature of any degenerative disease. At one stage or the other, cells most likely display calcium mishandling before their death, regardless of underlying primary pathological trigger.  Are all degenerative disorders similar to WS1?

I suggest a publication of this manuscript after very minor changes, mostly in concluding remarks.

Response

We sincerely thank the reviewer for their insightful feedback and suggestions, which have significantly improved the manuscript. Below are our responses to each of the points raised:

1. Figure 2 Y-axis Units and WFS1 Expression Data:
We appreciate the reviewer’s attention to detail. The units on the Y-axis in Figure 2 represent normalized expression levels. This information has now been added to the figure legend for clarity. Additionally, we clarified that the WFS1 expression data in Figure 2 were derived from brain tissue samples, specifically focusing on regions associated with neurodegeneration in both Wolfram Syndrome (WS1) and Alzheimer’s Disease (AD).

2. Figure 3 Y-axis Units and Visual System Expression:
We apologize for the oversight regarding the Y-axis units in Figure 3. These represent relative gene expression levels, and we have clarified this in the figure legend. Regarding the low level of WFS1 expression in the visual system, we agree that this observation is unexpected given the early degeneration of the visual system in WS1 patients. We have now included a discussion in the manuscript that addresses the potential complexity of WFS1’s role in neurodegeneration, suggesting that compensatory mechanisms or other molecular factors may influence the vulnerability of specific regions, which may not directly correlate with WFS1 expression levels.

3. Stronger Conclusion and Future Perspectives:
In line with the reviewer’s suggestion, we have strengthened the conclusion to more clearly address the similarities and differences between WS and AD. While both conditions exhibit shared pathological mechanisms, such as calcium mishandling and ER stress, we recognize that distinct cellular alterations may also contribute to their progression. The conclusion now discusses how WS models could be useful for AD research, particularly for testing therapies aimed at modulating calcium signaling or reducing ER stress. We also included a discussion on translational implications, noting that while treatments effective for one disorder might have potential for the other, more research is needed to confirm this hypothesis.

4. Calcium Mishandling in Degenerative Disorders:
In response to the reviewer’s broader question regarding the universality of calcium mishandling and ER stress across neurodegenerative diseases, we have added a new paragraph exploring this topic in the discussion. This section addresses how these mechanisms could be common features in various degenerative diseases, although the primary triggers and progression pathways may differ.

REFEREE N. 2

The authors put an extensive list of the potential mechanisms that the WFS1 gene can modulate the pathology underlying AD and psychiatric sleep disorders due to its critical role in calcium homeostasis and ER stress.  As such, WFS1 can serve as a promising therapeutic target in treating these disorders.

The content is informative and well organized. The review is easy to follow.

One minor point: The authors listed many ways the lack of WFS1 can potentially exacerbate the pathology of AD. Is there any evidence of WFS1 deficiency in AD?

Response

We are grateful for the reviewer’s positive feedback and for their helpful suggestion, which has enhanced the manuscript. Below is our response to the specific point raised:

1. Evidence of WFS1 Deficiency in AD:
We appreciate the reviewer’s insightful question regarding the evidence of WFS1 deficiency in AD. While direct evidence of WFS1 mutations in AD patients remains limited, we have included a discussion in the manuscript referencing recent studies that suggest WFS1 downregulation in AD-affected brain regions linked to ER stress and calcium homeostasis. These findings provide indirect evidence of WFS1’s involvement in AD pathology, and we have also noted that further research is needed to definitively establish whether WFS1 deficiency is a causative factor in AD progression.

We sincerely thank the reviewers for their valuable comments and constructive suggestions, which have greatly improved the quality and clarity of our manuscript. We believe that the revisions made in response to each comment have strengthened the scientific content and overall presentation. We appreciate the opportunity to address these insightful points and are confident that the manuscript now better highlights the relevance of WFS1 in both Wolfram Syndrome and Alzheimer’s Disease, as well as its potential as a therapeutic target.

Thank you once again for your time and consideration.

The Authors

Valerio Caruso and Luciana Rigoli

Reviewer 2 Report

Comments and Suggestions for Authors

The authors put an extensive list of the potential mechanisms that the WFS1 gene can modulate the pathology underlying AD and psychiatric sleep disorders due to its critical role in calcium homeostasis and ER stress.  As such, WFS1 can serve as a promising therapeutic target in treating these disorders.

The content is informative and well organized. The review is easy to follow.

One minor point: The authors listed many ways the lack of WFS1 can potentially exacerbate the pathology of AD. Is there any evidence of WFS1 deficiency in AD?

Author Response

(The authors gave the same response as above.)
